# Cascades Towards Noise-Induced Transitions on Networks Revealed Using Information Flows

**DOI:** 10.3390/e26121050

**Published:** 2024-12-04

**Authors:** Casper van Elteren, Rick Quax, Peter M. A. Sloot

**Affiliations:** 1Institute of Informatics, University of Amsterdam, 1098 XH Amsterdam, The Netherlands; r.quax@uva.nl (R.Q.); p.m.a.sloot@uva.nl (P.M.A.S.); 2Institute for Advanced Study, 1012 GC Amsterdam, The Netherlands; 3Complexity Science Hub Viennna, 1080 Vienna, Austria

**Keywords:** information theory, noise-induced transitions, metastability

## Abstract

Complex networks, from neuronal assemblies to social systems, can exhibit abrupt, system-wide transitions without external forcing. These endogenously generated “noise-induced transitions” emerge from the intricate interplay between network structure and local dynamics, yet their underlying mechanisms remain elusive. Our study unveils two critical roles that nodes play in catalyzing these transitions within dynamical networks governed by the Boltzmann–Gibbs distribution. We introduce the concept of “initiator nodes”, which absorb and propagate short-lived fluctuations, temporarily destabilizing their neighbors. This process initiates a domino effect, where the stability of a node inversely correlates with the number of destabilized neighbors required to tip it. As the system approaches a tipping point, we identify “stabilizer nodes” that encode the system’s long-term memory, ultimately reversing the domino effect and settling the network into a new stable attractor. Through targeted interventions, we demonstrate how these roles can be manipulated to either promote or inhibit systemic transitions. Our findings provide a novel framework for understanding and potentially controlling endogenously generated metastable behavior in complex networks. This approach opens new avenues for predicting and managing critical transitions in diverse fields, from neuroscience to social dynamics and beyond.

## 1. Introduction

Multistability, a fundamental characteristic of complex systems [1,2], describes the capacity of a system to occupy multiple stable states and transition between them. This phenomenon is ubiquitous, manifesting in diverse domains from neural networks [3,4] to opinion dynamics [5] and ecosystems [6]. While state transitions are often attributed to external perturbations, we propose a novel perspective: in networked systems, noise-induced transitions can occur endogenously. These transitions emerge from local interactions that cascade through the network, triggering large-scale regime shifts in a process we term the “domino effect”. This mechanism offers a new understanding of how complex systems can dramatically reconfigure without external forcing, challenging traditional views on system stability and change.

In nonlinear systems, such as interconnected neurons, noise plays a fundamental role in facilitating transitions between attractor states [7,8,9]. It enables the exploration of larger state spaces, allowing systems to escape local minima [10,11]. While multistability has historically been studied from an equilibrium perspective [10,12,13], recent research has revealed how network structure fundamentally affects the stability and transitions of complex systems [14,15,16,17].

Recent work has approached network control through algorithmic information theory, which measures the computational complexity of producing network states through controlled interventions [18,19,20]. While this provides powerful tools for steering networks through external manipulation, fundamental questions remain about how networks spontaneously transition between states through their internal dynamics. Our approach uses Shannon information theory to quantify the temporal correlations that emerge naturally as networks evolve, revealing how noise propagates through network structure to generate endogenous transitions. This complements algorithmic approaches by focusing on the statistical mechanisms underlying spontaneous state changes rather than the computational complexity of producing specific states.

Our study addresses a critical gap in understanding noise-induced transitions in networked dynamical systems out of equilibrium. We focus on systems where each node’s state evolves according to the Boltzmann–Gibbs distribution, a framework applicable to various phenomena including neural dynamics [21], opinion formation, and ferromagnetic spins [22]. An example of a noise-induced transition in this model executed on a network is shown in Figure 1.

We introduce two novel concepts: *initiator* nodes that propagate noise and destabilize the system, and *stabilizing* nodes that maintain metastable states. To quantify the impact of short-term and long-term correlations in these transitions, we propose two information-theoretic measures: integrated mutual information and asymptotic information. These metrics, computable from observational data, provide powerful tools for analyzing metastable dynamics across different time scales.

Integrated mutual information captures the transient destabilization of the system, revealing the role of initiator nodes in triggering systemic transitions. Asymptotic information, on the other hand, quantifies the long-term memory encoded by stabilizer nodes, which ultimately reverses the domino effect and settles the network into a new stable attractor. By manipulating these roles, we demonstrate how targeted interventions can either promote or inhibit systemic transitions, offering a new approach to controlling critical transitions in complex networks.

Our computational method uncovers a network percolation process that facilitates noise-induced transitions without external parameter changes, offering a fresh perspective on tipping points in complex networks [23,24,25,26]. This approach bridges the gap between local equilibrium dynamics and global system behavior, providing insights into how network structure influences systemic transitions [14,15,27,28,29].

By revealing the domino-like mechanisms of endogenous state transitions, our work has broad implications for predicting and potentially controlling critical transitions in diverse, complex systems. From enhancing brain plasticity to anticipating ecosystem shifts, this framework provides a foundation for understanding and managing multistability in an interconnected world.

## 2. Methods

Our study focuses on dynamical systems where the state transitions of individual nodes are governed by the Boltzmann–Gibbs distribution. This distribution, fundamental in statistical mechanics, provides a probabilistic framework for describing the behavior of systems in thermal equilibrium. In our context, it determines the likelihood of a node transitioning from one state to another based on the energy difference between states and a global noise parameter. Specifically, the probability of a node transitioning from state $s_{i}$ to state $s_{i}^{\prime }$ is given by:(1)$$P\left(s_{i}\to s_{i}^{\prime }\right)=\frac{1}{1+\exp(-\beta \Delta E\left(s_{i},s_{i}^{\prime }\right))},$$
where $\Delta E(s_{i},s_{i}^{\prime })$ represents the energy difference for the state transition, and β is the inverse temperature or noise parameter. This formulation captures the essence of how local interactions and global noise influence state changes in our networked system. Higher values of β correspond to lower noise levels, leading to more deterministic behavior, while lower β values introduce more randomness into the system’s dynamics. This framework allows us to model a wide range of phenomena, from neural activity to opinion dynamics, within a consistent mathematical structure.

Fluctuations and their correlations at time τ are captured using Shannon’s mutual information [30] shared between a node’s state ($s_{i}^{t}$) at time *t* and the entire future system state ($S^{t+\tau }$), $I(s_{i}^{\tau }:S^{\tau +t})$. The time lag *t* is used to analyze two key features of information flows of a system: the area under the curve (AUC) of short-term information and the sustained level of long-term information.

The contribution of a node to the dynamics of the system will differ depending on the network connectivity of a node (Figure A3) [31,32]. The total amount of fluctuations shared between the node’s current state and the system’s short-term future trajectory is computed as the integrated mutual information.
(2)$$\begin{array}{c} \mu \left(s_{i}\right)=\sum_{t=0}^{\infty }\left(I\left(s_{i}^{\tau }:S^{\tau +t}\right)-\omega_{\left(s_{i}\right)}\right)\Delta t. \end{array}$$

Intuitively, $\mu (s_{i})$ represents a combination of the intensity and duration of the short-term fluctuations on the (transient) system dynamics [31]. It reflects how much of the node state is in the “working memory” of the system.

The term $\omega \left(s_{i}\right)\in R_{\geq 0}$ represents the system’s long-term memory. As the system transitions between stable points, short-lived correlations evolve into longer-lasting ones, particularly among less dynamic nodes. When $\omega (s_{i})$ is positive, it indicates a separation of time scales: ephemeral correlations dissipate, giving way to slower, more persistent fluctuations. These enduring fluctuations reflect the multiple attractor states accessible to the system, with fewer dynamic nodes becoming more aligned with future system states.

Near a stable attractor, the system primarily generates short-lived fluctuations. However, as it approaches a tipping point, longer-lasting correlations emerge. These persistent correlations facilitate the system’s transition from one stable attractor to another, much like repeated nudges eventually push a ball over a hill. The asymptotic information, $\omega (s_{i})$, quantifies this transition potential. Higher values of $\omega (s_{i})$ indicate a greater likelihood of state transition, with the exact value reflecting each node’s contribution to the tipping behavior.

Asymptotic information distinguishes itself from other early warning signals—such as increased autocorrelation, critical slowing down captured by Fisher information, changes in skewness or kurtosis, and increased variance—by specifically measuring the system’s long-term memory and temporal correlation structure. While entropy captures the overall uncertainty or disorder in a system at a given moment, and mutual information quantifies the shared information between components at a particular time, asymptotic information focuses on the persistence of correlations over extended time periods. It reveals how past states influence future configurations, capturing aspects of the system’s dynamics that are not explained by instantaneous or short-term pairwise measures.

Using these information features, each node can be assigned to a different *role* based on their contribution to the metastable transition. We denote nodes with short-lived correlations as *initiators* pushing nodes towards a tipping point. In contrast, nodes with longer-lived correlations are referred to as *stabilizers*. For these nodes, their dynamics are less affected by short-lived correlations, and they require a higher mixing state to transition from one state to another. The role assignment will be further discussed in Section 3.5.

We compute information flows using exact calculations on a randomly generated connected graph of n=10 nodes. The states are grouped based on their distance to the tipping point, defined as the energy barrier between two locally stable states. For the Ising model, this corresponds to the collection of states where 〈S〉=0.5. We evaluate the conditional distribution up to τ=300 time steps.

This computational process scales exponentially with the number of nodes, $O\left(n\right)=2^{n}$, which limits its applicability to large-scale systems without employing variable reduction techniques such as coarse-graining. Extending this analysis to larger systems will be the focus of future research.

For detailed replication instructions, please refer to Appendix A.

## 3. Results

Our analysis reveals several key insights into the dynamics of metastable transitions and tipping points in complex networks. We observe a distinct *domino effect* where low-degree nodes initiate system destabilization. As the system approaches a tipping point, information flows shift from low-degree to high-degree nodes. We identify a rise in asymptotic information as a potential early warning signal for an impending tipping point. Finally, we uncover a division of roles among nodes, with some acting as *initiators* that propagate perturbations and others as *stabilizers* that influence the system’s transition between attractor states.

In Figure 2, we visualize the information flows at different stages as the system approaches the tipping point. While we present detailed analysis using the kite graph for simplicity, these findings generalize to other network structures, as demonstrated in Figure 3 and further elaborated in the Appendix A.

### 3.1. Information Flow Dynamics and the Domino Effect

To decompose the metastable transition, we consider local information flows in a given system partition, $S_{\gamma }=\left\{S^{\prime }\subseteq S|\langle S^{\prime }\rangle =\gamma \right\}$ where γ∈[0,1] represents the fraction of nodes that have state 1. This yields the conditional integrated mutual information:(3)$$\mu \left(s_{i}|\langle S\rangle \right)=\sum_{t=0}^{\infty }\left(I\left(s_{i}^{\tau }:S^{\tau +t}|\langle S^{\tau }\rangle \right)-\omega_{s_{i}}\right)\Delta t.$$

Details about the estimation procedure can be found in Section A.5.

Two key observations emerge from Figure 2:

First, the tipping point is reached through a domino effect, with low-degree nodes acting as initiators early in the process. These nodes, being more susceptible to noise (see Figure A3), are more likely to pass fluctuations to neighbors—akin to pushing a ball up a hill. Far from the tipping point (Figure 2a), lower-degree nodes show higher integrated mutual information, $\mu (s_{i}|\langle S\rangle )$, than higher-degree nodes. This noise injection by lower-degree nodes increases the likelihood of a metastable transition.

Second, an increase in asymptotic behavior corresponds to the system transitioning between attractor states. As shown in Figure 2b,c, asymptotic information remains low far from the tipping point and steadily increases as the system approaches it. Nodes with higher asymptotic information possess greater predictive power regarding which side of the tipping point the system will settle on.

### 3.2. Path Analysis and Tipping Point Trajectories

To illustrate the information encoded in these flows, we computed trajectories from the attractor state S={0,⋯,0}, simulated for t=5 steps. Figure 4 shows a trajectory that maximizes:$$\log p\left(S^{t+1}|S^{t},S^{0}=\left\{0,\cdots ,0\right\},\langle S^{5}\rangle =0.5\right).$$

These trajectories reveal how the information flows measured in Figure 2c are generated by the sequence of flips originating from the tail of the kite graph. Tail nodes are uniquely positioned to pass on fluctuations to their neighbors, eventually causing a cascade of flips that reach the tipping point. This simple example illustrates how the network structure can influence the system’s dynamics and the information flows that precede a metastable transition. Where noise pushes the system towards a tipping point, originating first in low-degree nodes for dynamics governed by the Boltzmann–Gibbs distribution.

### 3.3. Network Structure and Node Roles in Metastable Transitions

The domino effect is not solely determined by node degree. As the system nears the tipping point, network effects become significant. For instance, in the kite graph, node 8 (degree 2) exhibits the highest integrated mutual information when 2 bits are flipped (Figure 2b). In contrast, node 3 (degree 6) shows low shared information prior to the tipping point but high shared information at the tipping point.

This transition highlights how the network structure as a whole contributes to a system’s behavior. Local structural measures, such as degree centrality, may undervalue a node’s contribution towards a tipping point and the eventual settlement in a new attractor.

### 3.4. Tipping Point Dynamics and Information Flow

At the tipping point, the system is most likely to either move to a new attractor state or relax back to its original state (Figure 4). Path analysis reveals that the most likely paths to the tipping point result in a configuration where a high-degree cluster of nodes must flip. This trajectory is less likely than reversing the path shown in Figure 4, explaining why most tipping points “fail” and relax back to the original attractor state (Figure 5b).

The increased information of node 8 around the tipping point can be understood by considering its predictive power about the system’s future. As shown in Figure 5a, both node 3 and node 8 have low uncertainty about the future system state, but the nature of this certainty differs. Node 3 is more certain that the average system state will equal its state at the tipping point, while node 8 is more certain that the future system state will have the opposite sign to its state at the tipping point.

### 3.5. Role Division and Interventions in Tipping Behavior

We approximate the role of a node *i* using the difference between integrated mutual information and asymptotic information:(4)$$r_{i}=\max_{\langle S\rangle }\mu^{*}\left(s_{i}|\langle S\rangle \right)-\max_{\langle S\rangle }\omega^{*}\left(s_{i}\right)\in \left[-1,1\right],$$
where $\mu^{*}$ and $\omega^{*}$ are normalized versions of μ and ω, respectively.

Nodes with role values close to 1 are classified as “initiators” with high predictive information about short-lived system trajectories. Nodes with values close to −1 are “stabilizers” with high long-term predictive information about future system states.

We validated these roles using simulated interventions (Figure 3). Pinning initiator nodes to the 0 state promotes tipping points while pinning stabilizer nodes is essential for stabilizing transitions between attractor states.

## 4. Discussion

Understanding how metastable transitions occur may help in understanding how, for example, a pandemic occurs or a system undergoes critical failure. In this paper, dynamical networks governed by the Boltzmann–Gibbs distribution were used to study how endogenously generated metastable transitions occur. The external noise parameter (temperature) was fixed such that the statistical complexity of the system behavior was maximized (see Section A.2).

The results show that in the network, two distinct node types could be identified: initiator and stabilizer nodes. Initiator nodes are essential early in the metastable transition. Due to their high degree of freedom, these nodes are more affected by external noise. They are instigators and propagate noise in the system, destabilizing more stable nodes. In contrast, stabilizer nodes have a low degree of freedom and require more energy to change state. These nodes are essential for the metastable behavior as they stabilize the system macrostate. During the metastable transition, a domino sequence of node state changes is propagated in an ordered sequence toward the tipping point.

This domino effect was revealed through two information features, unveiling an information cascade underpinning the trajectories toward the tipping point.

Integrated mutual information captured how short-lived correlations are passed on from the initiator nodes. In the stable regime (close to the ground state), low-degree nodes drive the system dynamics. Low-degree nodes destabilize the system, pushing the system closer to the tipping point. In most cases, the initiator nodes will fail to propagate the noise to their neighbors. On rare occasions, however, the cascade is propagated progressively from low degree to higher and higher degree. A similar domino mechanism was recently found in climate science [6,27]. Wunderling and colleagues provided a simplified model of the climate system, analyzing how various components contribute to the stability of the climate. They found that interactions generally stabilize the system dynamics. If, however, a metastable transition was initialized, the noise was propagated through a similar mechanism as found here, i.e., an initializer node propagated noise through the system, which created a domino effect that percolated through the system.

An increase in asymptotic information forms an indicator of how close the system is to a tipping point. Close to the ground state, the asymptotic information is low, reflecting how transient noise perturbations are not amplified, and the system macrostate relaxes back to the ground state. As the system approaches the tipping point, the asymptotic information increases. As the distance to the ground state increases, the system is more likely to transition between metastable states. After the transition, there remains a longer-term correlation. Asymptotic information reflects the long(er) timescale dynamics of the system. This “rest” information peaks at the tipping point as the system chooses its next state.

The information viewpoint uniquely offers an alternative view to understand how metastable transitions are generated by dynamical networks. Two information features were introduced that decompose the metastable transition in sources of high information processing (integrated mutual information) and distance of the system to the tipping point (asymptotic information). A domino effect was revealed, whereby low-degree nodes initiate the tipping point, making it more likely for higher-degree nodes to tip. On the tipping point, long-term correlations stabilize the system inside the new metastable state. Importantly, the information perspective allows for estimating integrated mutual information directly from data without knowing the mechanisms that drive the tipping behavior. The results highlight how short-lived correlations are essential to initiate the information cascade for crossing a tipping point.

## 5. Conclusions

Our information-theoretic approach offers an alternative view to understanding *how* metastable transitions are generated by dynamical networks. Two information features were introduced that decompose the metastable transition in sources of high information processing (integrated mutual information) and distance of the system to the tipping point (asymptotic information). A domino effect was revealed, whereby low-degree nodes initiate the tipping point, making it more likely for higher-degree nodes to tip. On the tipping point, long-term correlations stabilize the system inside the new metastable state. Importantly, the information perspective allows for estimating integrated mutual information directly from data without knowing the mechanisms that drive the tipping behavior. The results highlight how short-lived correlations are essential to initiate the information cascade for crossing a tipping point.

## 6. Limitations

Integrated mutual information was computed based on exact information flows. This means that for binary systems, it is necessary to compute a transfer matrix on the order of $2^{\left|S\right|}\times 2^{\left|S\right|}$. This reduced the present analysis to smaller graphs. It would be possible to use Monte-Carlo methods to estimate the information flows. However, $I(s_{i}^{\tau }:S^{\tau +t})$ remains expensive to compute. When using computational models, it is necessary to compute the conditional and marginal distributions, which are on order $O(2^{\left|S\right|})$ and $O(2^{t|S|})$, respectively. In Section A.11, we give a proof of principle of how the results presented here would generalize to larger systems.

In addition, the decomposition of the metastable transition depends on the partition of the state space. Information flows are, in essence, statistical dependencies among random variables. Here, the effect of how the tipping point was reached was studied by partitioning the average system state in terms of the number of bits flipped. This partitioning assumes that the majority of states prior to the tipping point are reached by having fraction c∈[0,1] bits flipped. The contribution of each system state over time, however, reflects a distribution of different states; reaching the tipping point from the ground state 0 can be done at t−2 prior to tipping by either remaining in 0.4 bits or transitioning from 0.3 bits flipped to 0.4 and eventually to 0.5 in 2 time steps. The effect of these additional paths showed marginal effects on the integrated mutual information and asymptotic information.

Information flows conditioned on a partition is a form of conditional mutual information [33]. Prior results showed that conditional information produces synergy, i.e., information that is only present in the joint of all variables but cannot be found in any of the subsets of each variable. Unfortunately, there is no generally agreed-upon definition of how to measure synergy [34,35], and different estimates exist that may over or underestimate the synergetic effects. By partitioning, one can create synergy as, for a given partition, each spin has some additional information about the other spins. For example, by taking the states such that 〈S〉=0.1, each spin “knows” that the average of the system equals 0.1. This creates shared information among the spins. Analyses were performed to estimate synergy using the redundancy estimation $I_{min}$[36]. Using this approach, no synergy was measured that affected the outcome of this study. However, it should be emphasized that synergetic effects may influence the causal interpretation of the approach presented here.

A general class of systems was studied governed by the Boltzmann–Gibbs distribution. For practical purposes, the kinetic Ising model was only tested, but we speculate that the results should hold (in principle) for other systems dictated by the Boltzmann–Gibbs distribution. We leave the extension to other Hamiltonian systems for future work.

The practical implementation of interventions based on our theoretical framework faces several real-world challenges. First, in actual complex systems, measuring and monitoring the complete state space in real time may be technically infeasible or prohibitively expensive. Second, the ability to perform precise, targeted interventions on specific components of the system may be limited by physical constraints or technological capabilities. Third, the assumption of perfect knowledge about system parameters and state transitions may not hold in real-world scenarios where noise, measurement errors, and external perturbations are present. Furthermore, the time scales at which interventions need to be implemented may be too rapid for practical human or automated response systems. These practical limitations suggest that while our framework provides valuable theoretical insights, its application may require significant adaptation and simplification for real-world implementation, potentially trading off optimal control for practical feasibility.

## Figures and Tables

**Figure 1 entropy-26-01050-f001:**
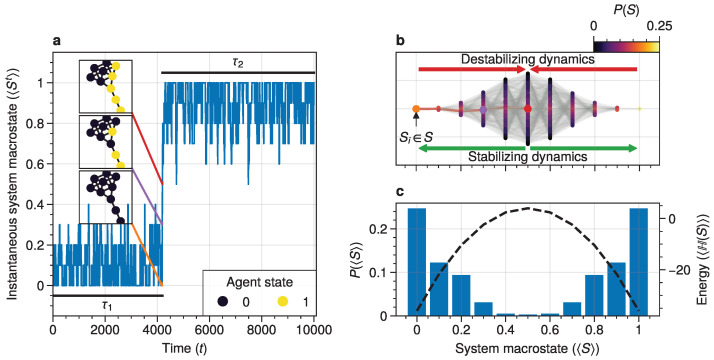
A dynamical network governed by kinetic Ising dynamics produces multistable behavior. (**a**) A typical trajectory is shown for a kite network for which each node is governed by the Ising dynamics with β≈0.534. The panels show system configurations $S_{i}\in S$ as the system approaches the tipping point (orange to purple to red). For the system to transition between attractor states, it must cross an energy barrier (**c**). (**b**) The dynamics of the system can be represented as a graph. Each node represents a system configuration $S_{i}\in S$ such as depicted in (**a**). The probability for a particular system configuration p(S) is indicated with a color; some states are more likely than others. The trajectory from (**a**) is visualized. Dynamics that move towards the tipping point (midline) destabilize the system, whereas moving away from the tipping point are stabilizing dynamics. (**c**) The stationary distribution of the system is bistable. Crossing the tipping point requires crossing a high-energy state (dashed line). Transitions between the attractor states are infrequent and rare. For more information on the numerical simulations, see Section A.2.

**Figure 2 entropy-26-01050-f002:**
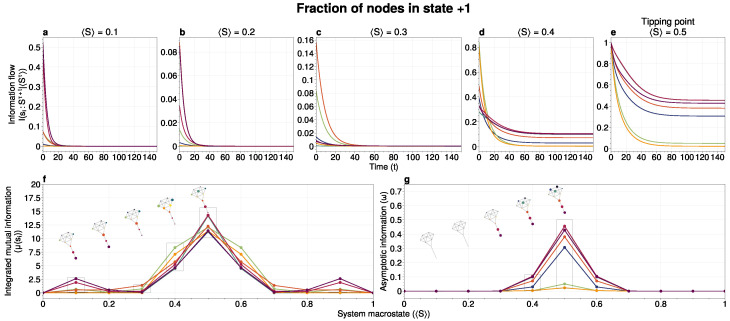
(**a**–**e**) Information flows as distance to a tipping point, where each line color corresponds to the matching-colored node in the kite graph inset. Far away from the tipping point, most information processing occurs in low-degree nodes (colored in blue/purple, tail of kite). As the system moves towards the tipping point, the information flows increase and shift towards higher-degree nodes (colored in red/orange, core of kite). (**f**) Integrated mutual information as a function of distance to the tipping point. The graphical inset plots show how noise is introduced far away from the tipping point in the tail of the kite graph (blue/purple nodes). As the system approaches the tipping point, the local information dynamics move from the tail to the core of the kite (red/orange nodes). (**g**) A rise in asymptotic information indicates that the system is close to a tipping point. At the tipping point, the decay maximizes as trajectories stabilize into one of the two attractor states. The color of each line consistently matches its corresponding node in the kite graph visualization.

**Figure 3 entropy-26-01050-f003:**
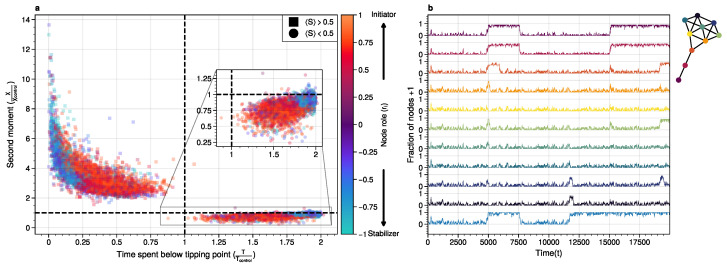
For system to cross a tipping point, two distinct types of nodes are essential: **stabilizers**, which contain information about the system’s next attractor state and facilitate transitions between states; and **initiators**, which propagate noise through the system. (**a**) The effect of causal pinning interventions on node 0 states in Erdős–Rényi graphs (N=100, 10 nodes each, p=0.2, 6 seeds) is shown. Normalized system fluctuations (second moment) and time spent below the tipping point relative to the control are presented per network to indicate the effect of the pinning interventions. Pinning initiators increase tipping points while pinning stabilizers prevent tipping and increase noise above the tipping point. For more details on role approximation, see Section 3.5. (**b**) To exemplify the effect of the causal interventions in (**a**), typical system trajectories underpinning interventions on a node for the kite graph are shown. Colors reflect intervention on corresponding nodes in the inset kite graph. Initiator-based interventions remove fluctuations below the tipping point (<0.5) and increase fluctuations above, whereas stabilizer-based interventions stabilize tipping points while increasing noise.

**Figure 4 entropy-26-01050-f004:**
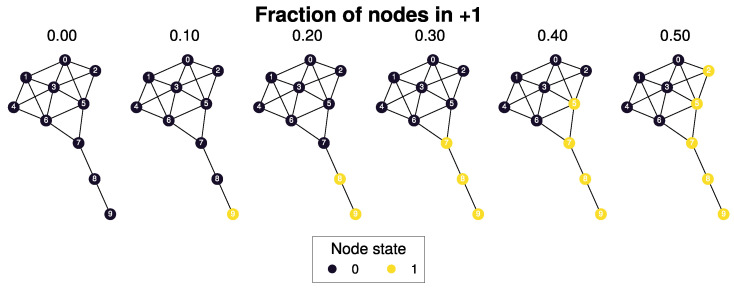
The tipping point is initiated from the bottom up. Each node is colored according to state 0 (black) and state 1 (yellow) Shown is a trajectory towards the tipping point that maximizes $\sum_{t=1}^{5}\log p\left(S^{t+1}|S^{t},S^{0}=\left\{0\right\},\langle S^{5}\rangle \right)=0.5)$. As the system approaches the tipping point, low-degree nodes flip first and recruit “higher” degree nodes to further destabilize the system and push it towards a tipping point. In total, 30,240 trajectories reach the tipping point in 5 steps, and 10 trajectories have the same maximized values as the trajectory shown in this figure (see Figure A7 for the remaining trajectories and probabilities).

**Figure 5 entropy-26-01050-f005:**
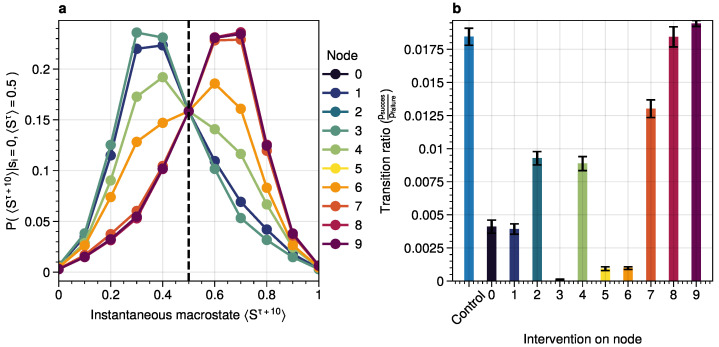
(**a**) Shown are the conditional probabilities at time t=10 relative to the tipping point. The shared information between the hub node 3 and the tail node 8 is similar but, importantly caused through different sources. The hub (node 3) has high certainty that the system macrostate will be the same sign as its state. In contrast, node 8 has high certainty that the system macrostate will be opposite to its state at the tipping point. This is caused by the interaction between the network structure and the system dynamics whereby the most likely trajectories to the tipping point from the stable regime are mediated by the noise-induced dynamics from the tail to the core in the kite graph (see main text). (**b**) Successful metastable transitions are affected by network structure. Successful metastable transitions are those for which the sign of the macrostate is not the same prior to and after the tipping point, e.g., the system going from the 0 macrostate side to the +1 macrostate side or vice versa. Shown here are the number of successful metastable transitions for Figure 3 under control and pinning interventions on the nodes in the kite graph.

## Data Availability

The datasets generated and/or analyzed during the current study are available in the https://github.com/cvanelteren/metastability repository, accessed date (28 November 2024).

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
