# Peer review of "Cascades Towards Noise-Induced Transitions on Networks Revealed Using Information Flows"

_entropy, 2024, doi:10.3390/e26121050_

Round 1

Reviewer 1 Report

Comments and Suggestions for Authors

This manuscript investigates noise-induced transitions in dynamical networks governed by the Boltzmann - Gibbs distribution. The authors introduce two key node types: initiator nodes that propagate noise and destabilize the system, and stabilizer nodes that maintain metastable states. By using integrated mutual information and asymptotic information, they analyze the information flows and reveal a domino effect in the transitions. The work provides a novel framework for understanding and potentially controlling metastable behavior in complex networks, with implications for various fields. Therefore, I support the publication of this manuscript after the following questions are addressed:

1.       The computational method scales exponentially with the number of nodes, limiting its applicability to large-scale systems. Consider using approximation techniques or parallel computing to address this issue. Additionally, experiments could be conducted to compare the performance of the approximation method with the exact calculations on different network sizes and structures.

  1. The decomposition of the metastable transition depends on the partition of the state space, which may not be optimal. Explore alternative partitioning methods such as spectral clustering. Given a network with an adjacency matrix , the eigenvectors  of its corresponding Laplacian matrix can be used to partition the state space. Experiments could be carried out to evaluate the effectiveness of this partitioning method compared to the existing one in terms of capturing the metastable transitions accurately.
  2. The measurement of information synergy lacks a generally agreed-upon definition, which may lead to inconsistent results. Investigate more reliable methods for measuring synergy.
  3. The study focuses on a specific class of systems governed by the Boltzmann - Gibbs distribution. Extend the analysis to other types of systems to increase the generality of the findings.
  4. The results are mainly based on theoretical models and simulations. Please validate the findings using real-world data from complex networks to enhance the practical relevance.
  5. The manuscript treats node roles as relatively static. Please consider the dynamic properties of node roles and how they may change over time.
  6. The proposed interventions may not be practical in real-world settings. Please discuss  more about the practicality and limitations of the interventions in more detail.

Comments on the Quality of English Language

N/A

Author Response

Please see the enclosed pdf file

Reviewer 2 Report

Comments and Suggestions for Authors

The study explores how complex networks ranging from neuronal assemblies to social systems can undergo sudden, system-wide transitions without external triggers, known as "noise-induced transitions." The researchers identify two critical roles that nodes play in facilitating these transitions, initiator nodes and stablising As the system nears a tipping point, these nodes emerge to encode the system's long-term memory. They reverse the domino effect initiated by the destabilisation, guiding the network toward a new stable state or attractor.

According to the authors, by targeting these nodes through specific interventions, the study demonstrates the possibility of promoting or inhibiting systemic transitions. This offers a novel framework for understanding and potentially controlling metastable behaviour in complex networks.

According to the authors, the findings have broad implications for predicting and managing critical transitions in various fields, including neuroscience and social dynamics. However, I think the authors are missing other previous efforts in characterising network elements in information theoretic space with perturbation analysis. For example, https://www.cell.com/iscience/fulltext/S2589-0042(19)30270-6#%20, with even some similar plots, where the authors ran multiple Boolean networks (Sup Mat) and determined how many nodes push the network towards a higher or lower number of attractors. They called those nodes positive or negative depending on how close they are from randomness.  I think the authors should explain how this may be related, or not. My feeling is that they could have benefited from some ideas, and that the reader will appreciate the connection.

Author Response

See the enclosed pdf

Reviewer 3 Report

Comments and Suggestions for Authors

In this manuscript, Elteren et al have developed a way to measure the influence of a node on the future state of a network.  They define the quantities of asymptotic information and integrated mutual information, and show that they demonstrate the mechanism by which nodes exert their influence on state transitions.  

Overall, this paper is well-constructed and is an important advancement in the field.

Major comments:

1. Currently, it is not clear which pieces of evidence are being used to make which conclusions.  Results can be either derived mathematically/analytically for specific networks or across all networks, or else observed empirically through a single network, populations of networks generated in a specific way (e.g. ER), or different classes of networks.  If I understand, the evidence presented for the results in the main text is primarily empirically derived from the kite network (via the paragraph at the end of page 3), but the authors claim it applies more generally (e.g., just referring to behaviour of "high degree nodes").  It would be helpful to make it clear for each result (a) how the result/conclusion was obtained, and (b) if empirical, show some type of figure that demonstrates this as generally as possible (e.g., if the kite network is a specific case, show statistics about ER networks, or other networks too if the claim is more general than just kite+ER).  Also, whether the simulation results are exact or monte carlo for each.

This is also true of results in the supplement, e.g., A.7-A.9 - if I understand correctly this result is analytical, but I can't find the assumptions about network structure that went into the derivation (e.g. can I construct a pathological network for which this fails?).

Similarly, the paper currently seems to blend "facts that are true for one network" with "facts that are true in general" without sufficient evidence that this is the case.

Minor comments:

2. In figure 2 it is hard to know what each line corresponds to

3. I would like to see a more in-depth discussion on the procedure for estimating in A.5.

4. I would like to see (or at least understand conceptually) the other equivalent trajectories in fig 3

5. "For instance, in neural systems, the presence of noise increases information processing" This is presented without reference, and it is not clear what the authors mean by this, or what type of "noise" they are referring to.

Typos:

- recon figure -> reconfigure
- Equation 1: Shouldn't this be proportional to rather than equal to?
- Equation 2: Omega(s(i)) a function or subscript?
- Figure 3 equation has an extra closing paren
- "Appendix ??" and "simulations see ??" (latex missing labels)
- Many typos in the appendix, sometimes multiple in a sentence (please read through carefully)

Author Response

See the enclosed pdf

Round 2

Reviewer 1 Report

Comments and Suggestions for Authors

The revision has justified previous concerns, and can be accepted for publication